



# Impact of climate and hydrochemistry on shape variation – a case study on Neotropical cytheroidean Ostracoda

Claudia Wrozyna[1], Thomas A. Neubauer[2,3], Juliane Meyer[1], Maria Ines F. Ramos[4], Werner E. Piller[1]

[1] Institute of Earth Sciences, NAWI Graz Geocenter, University of Graz, Graz, 8010, Austria
[2] Department of Animal Ecology & Systematics, Justus Liebig University, Giessen, 35392, Germany
[3] Naturalis Biodiversity Center, Leiden, 2300 RA, The Netherlands
[4] Coordenação de Ciências da Terra e Ecologia, Museu Paraense Emílio Goeldi, 66077-830, Brazil

Correspondence to: Claudia Wrozyna (claudia.wrozyna@uni-graz.at)

**Abstract.** How environmental change affects a species' phenotype is crucial not only for taxonomy and biodiversity assessments but also for their application as (paleo-)ecological indicators. Previous investigations addressing the impact of climate and hydrochemical regime on ostracod valve morphology have yielded quite contrasting results. Frequently identified ecological factors influencing carapace shape are salinity, cation and sulphate concentrations and alkalinity. Here, we present a thorough approach integrating data from carapace outline and surface details of the ubiquitous Neotropical cytheroidean ostracod species *Cytheridella ilosvayi*, as well as several climatic and hydrochemical variables, in order to investigate a potential link between morphology and environmental conditions. A previous study lately demonstrated considerable biogeographical variation in valve morphology among Floridian, Mexican and Brazilian populations of this species. We hypothesize that the climatic differences between the regions it inhabits and associated differences in hydrochemical regimes have influenced valve morphology and eventually led to biogeographically distinctive groups. Generalized least-squares Procrustes Analyses based on outline and structural features were applied to left and right valves of adult females and males. The analyses identified relative carapace length and shape symmetry as most important morphological characteristics representing shape change across all datasets. Two-block partial least-squares analyses and multiple regressions indicate strong relationships between morphological and environmental variables, specifically with temperature seasonality, annual precipitation and chloride and sulphate concentrations. We hypothesize that increased temperature seasonality slowed down growth rates during colder months, potentially triggering the development of shortened valves with well-developed brood pouches. Differences in chloride and sulphate concentrations, related to fluctuations in precipitation, are considered to affect valve development via controlling osmoregulation and carapace calcification. These factors represent hitherto unknown drivers for ostracod ecophenotypy and emphasise that environmental predictors for morphological variability are not consistent across non-marine ostracods.





## 1 Introduction

Understanding how species respond to environmental change is crucial for their application as proxies for past climate fluctuations as well as forecasting future dynamics and distribution of species. Morphological diversity represents a key

character for the interpretation of faunal changes (Wagner and Erwin, 2006) and ecological shifts (Mahler et al., 2010) and urges discussions about speciation and extinction processes through time (e.g., Ciampaglio, 2004). Differences in shape and size among species have been shown to relate with changes of environmental parameter, in particular, differences in temperature across various clades (e.g., Loehr et al., 2010; Maan and Seehausen, 2011; Danner and Greenberg, 2015). Within freshwater invertebrates, ecophenotypic response has been documented for a variety of species, both recent and fossil (e.g.,

Hellberg et al., 2001; Zieritz and Aldridge, 2009; Inoue et al., 2013; Neubauer et al., 2013; Clewing et al., 2015).

Ostracods represent a model group for the study of ecophenotypical variation in response to environmental change (Anadón et al., 2002; Frenzel et al., 2012; Fürstenberg et al., 2015; van der Meeren et al., 2010). Due to their calcitic valves, they have an excellent fossil record and are utilized as palaeoenvironmental and biostratigraphic indicators (Anadón et al., 2002). A number of studies has shown that ornamentation, noding, sieve pore shape, and carapace size are linked to environmental

factors, e.g., salinity, temperature, water depth and nutrient availability (van Harten, 1975; Yin et al., 1999; Majoran et al., 2000; van Harten, 2000; Anadón et al., 2002; Frenzel and Boomer, 2005; Medley et al., 2007; Marco-Barba et al., 2013; Meyer et al., 2016; Boomer et al., 2017). Especially with the rise of morphometric techniques, investigations also dealing with carapace shape variation in relation to environmental variables have increased (Yin et al., 1999; Baltanas et al., 2002; Baltanas et al., 2003; van der Meeren et al., 2010; Ramos et al., 2017; Grossi et al., 2017). Yet, the use of morphological data, even

those based on morphometric analyses (Baltanas et al., 2002; Baltanas et al., 2003; van der Meeren et al., 2010; Grossi et al., 2017), has been restricted to either landmark-based or outline-based studies but have rarely used a combination of both (e.g., Ramos et al., 2017). Few studies integrate geographic gradients into their statistical analyses and corresponding climate variables or a reduced number of predictor variables. Moreover, shape-environment relationships are commonly identified based on simple linear regressions or qualitative observations on multivariate ordination methods.


Here, we apply a thorough approach integrating data from carapace outline and surface details, as well as several climatic and hydrochemical variables, in order to investigate a potential link between morphology and environmental conditions. Subject of study are valves of the Neotropical cytheroidean ostracod species *Cytheridella ilosvayi* Daday, 1905. Wrozyna et al. (2016) and Wrozyna et al., (under review) lately demonstrated considerable biogeographical variation in valve morphology among

Floridian, Mexican and Brazilian populations of that species. Morphological differences in populations of *C. ilosvayi* are discernible for both valves and appendages, for adult and juvenile (A-1 to A-3) stages and across sexes, suggesting that morphological divergence is a result of long-term biogeographic isolation (Wrozyna et al. submitted). While the morphological



aspects of the biogeographic variability in *C. ilosvayi* are well understood, the causes for the regional differences have not been investigated. So far, is the knowledge of ecophenotypical characteristics of ostracods restricted to few species and/or few environmental variables. Moreover, the limited use of modern analytical methods and insufficient environmental data allowing thorough representation of natural variability has confined the possibility to identify the fundamental drivers of shape variation.

We hypothesize that the climatic differences between the regions inhabited by *Cytheridella ilosvayi* and associated differences in hydrochemical regimes have influenced valve morphology and finally led to biogeographically distinctive groups. We apply two-block partial least squares analyses and multiple regression analyses, in order to test for covariation between the two sets of parameters (morphology, environment) and to identify the morphological characteristics and environmental variables that contribute most to the relationship.

## 2 Material and Methods

### 2.1 Material

Specimens of *C. ilosvayi* derive from several sampling campaigns in Florida, Mexico and Brazil during 2009–2015 (Fig. 1). A detailed list of the sampled localities is available in Appendix S1. Only adult valves were utilized in this study, providing a

sufficient number of left and right valves across both sexes. Right and left valves were investigated separately due to dimorphism in size and shape (Wrozyna et al., 2014). Beyond that, females and males were analyzed separately because a large part of within-valve variation has been shown to depend on sexual differences (such as the presence of brood pouches in females; Wrozyna et al., 2016).

### 2.2 Predictor variables

Altogether 15 variables were included in the analyses. Simultaneously to water sampling, field variables (electrical conductivity, water temperature and pH) were measured *in situ* at all sample sites using a WTW multi-sensor probe (Multi 3420 Set C). Water samples were promptly filtrated using a syringe filter with a filter pore size of 0.45 µm and stored until analysis. Major ions were measured at the laboratory center of Joanneum Research in Graz by ion chromatography (Dionex ICS-3000). As the variables measured per sampling station only provide a snapshot of the local ecological conditions, the set

of variables was supplemented with bioclimatic data from the WorldCLIM database (WorldClim, 2017), providing data on monthly to yearly scales. From the many variables available we included annual mean temperature [°C] (BIO1), mean diurnal range [°C] (BIO2; mean of monthly maximum-minimum temperature), temperature seasonality [°C] (BIO4; standard deviation *100), annual precipitation [mm] (BIO12) and precipitation seasonality (BIO15; coefficient of variation), each with a spatial resolution of 30″. We chose not to include all bioclimatic variables because many of them are highly correlated, causing issues





for the regressions. Bioclimatic variables and occurrence data were linked in ESRI ArcGIS v. 10.4 with the tool "Extract Multi Values to Points". Environmental variables are provided in Appendix S2.

### 2.3 Methods

#### 2.3.1 Generalized Procrustes Analysis

Two main approaches are available for geometric morphometric analyses, one focusing on point-data, one on outlines and surfaces. Landmarks allow to study shape variations via a configuration of clearly defined homologous points in 2D or 3D space (Bookstein, 1996; Webster and Sheets, 2010). Outline-based morphometric analyses, on the other hand, deal with open or closed curves or curve segments and largely neglect information from homologous points (Kuhl and Girdina, 1982;

Lohmann, 1983; Haines and Crampton, 2000; Sheets et al., 2006). The majority of morphometric analyses of ostracods has focused on outline methods (e.g., Danielopol et al., 2008; Tanaka, 2009; Namiotko et al., 2012; Gitter et al., 2015) mainly because of lacking homologous points in most non-marine taxa (Baltanas et al., 2002). To overcome the insufficiency of addressing only a part of the morphological spectrum the method of sliding semilandmarks was developed (Bookstein, 1991; 1997; Gunz and Mitteroecker, 2013). Semilandmarks offer a convenient way to quantify two- or three-dimensional

homologous curves and surfaces, and to analyze them together with traditional landmarks (Gunz and Mitteroecker, 2013).
In our case, valve morphology was captured using a combination of landmarks and semilandmarks. Eight points were chosen as landmarks (LM): anterior pore tubuli (LM 1–5, type-I) and the dorsal dip point of the posterior curvature (LM 6, type-II), as well as to delimitate maximum anterior and posterior curvatures (LM 7–8, type-III). Carapace outline was defined by two curves between LM 7 and 8, each comprising 30 equidistantly spaced semilandmarks (Fig. 2; see also Wrozyna et al., 2016).

All points were set on digitized SEM images using the program TpsDig v. 2.17 (Rohlf, 2013). The sliders file determining sliding direction of the semilandmarks during the Procrustes alignment was created in TpsUtil v. 1.58 (Rohlf, 2015). A generalized least-squares Procrustes Analysis, computing consensus configuration, partial warps and relative warps, was performed in TpsRelw v. 1.65 (Rohlf, 2016) Thin-plate spline deformation grids were used to visualize deviations of selected configurations from the mean and to identify morphological characteristics that account for differences among geographic

regions. For details on the method see (Rohlf and Slice, 1990) and (Bookstein, 1996).
We ran preliminary analyses for each dataset to identify major outliers that may bias the morphometric analyses by overemphasizing particular directions in the morphospace (and associated morphological characteristics). Such distortion may severely impede sound interpretation of follow-up statistical analyses.

#### 2.3.2 Statistics

In order to study the covariance between shape variation and environmental variables, two-block partial least-squares analyses (PLS) were performed using software PAST 3.14 (Hammer et al., 2001). As a great advantage over other ordination methods





such as principal components analysis, this method disregards within-block variation that may mask between-block covariance (Mitteroecker and Bookstein, 2011, 2008). Using all relative warps in the PLS might severely bias the pattern, because – contrary to their descending significance in terms of explaining shape variation – they would be treated equally by the analysis. Therefore, we restricted the morphological block to RW 1–20, which account for at least 98.6% of the total shape variation in

all four datasets. The environmental variables were log10-transformed to constrain the orders of magnitude involved. PLS was computed based on the covariance matrix.

To detect which parameters contribute to shape variation, multiple regression analyses were conducted on selected relative warps in the statistical environment R v. 3.3.2 (R Core Team, 2016). Only warps 1) along which biogeographic differentiation was observed, 2) with an amount of shape variation higher than 10% of the total variation, and 3) those yielding high loading

values in the PLS analysis were considered. These selection criteria were chosen in order to prevent from misinterpreting seemingly strong relationships between shape and environmental variables. Since the environmental parameters are likely to be highly correlated, eventual models including all variables might be strongly skewed and susceptible to misinterpretation. Therefore, we employed a stepwise selection of variables based on the variance inflation factor (VIF), which is an estimator of multicollinearity among variables (Quinn and Keough, 2002). As a rule of thumb, VIF values greater than ten indicate the

presence of multicollinearity (Quinn and Keough, 2002); some authors even consider values above five evidence of collinearity (Heiberger and Holland, 2004). The applied function iteratively removes collinear variables by calculating the VIF of variables against each other (for the script see (Ijaz, 2013); R package 'fmsb' v. 0.5.2 (Nakazawa, 2015) is required for this procedure. VIF values were calculated with package 'HH' v. 3.1-32 (Heiberger, 2016). To enhance the models further, multiple regressions using backward stepwise selection by evaluation of the Akaike Information Criterion (AIC) were performed with

the remaining set of factors.

Normality of model residuals was tested with Shapiro-Wilk tests. In case normality was not achieved, residual distributions were assessed qualitatively using Q-Q-plots; only if the majority of cases match the expected distribution, a model was considered significant. Finally, we used the R package 'hier.part' v. 1.0-4 (Walsh and Mac Nally, 2013) to evaluate the independent contribution of each predictor to the (reduced) models.

**3 Results**

The relative warps analysis yielded different results for males and females, while patterns were largely consistent within sexes (Fig. 3, 4). Along the first three relative warps (RW), Mexican females have little overlap with Brazilian/Floridian ones. Only some of the specimens from Punta Laguna in northern Yucatan seem to be morphologically closer to the Floridian group and cluster apart in the analyses of both valves. Brazilian and Floridian individuals have a distinctly higher overlap and differentiate

only little along RW 2. A clear differentiation within both clusters, like in the Mexican group, is lacking. Group differentiation in male valves is quite contrary: Floridian specimens have little overlap with Brazilian ones in both valves along RW 1, while Mexican specimens are hardly separable from either group along any of the first 3 RWs. However, the differentiation between



some Punta Laguna valves and remaining Mexican carapaces along RW 1 is comparable to the patterns observed for females. Mexican and Brazilian males show slight biogeographic differentiation along RW 2 (left valves) and RW 3 (right valves), respectively. No clustering is observed for higher warps in either sex or valve.

Similar to the patterns posed by the scatter plots, shape variation along RWs is largely consistent within valves but differs

slightly between sexes. (Here we display only axes along which biogeographic discrimination is observed. See Wrozyna et al. 2016 for within-group variation) The most important morphological characteristic representing shape change along RW 1 in both females and males and right and left valves, is relative carapace length (Fig. 3, 4). However, the exact expression differs between sexes: valve outline in males varies between elongate-elliptical and short-asymmetrical with slightly inflated anterior part, and between elongate-elliptical and short-asymmetrical with distinctly inflated posterior region (i.e., brood pouch) in

females. In addition to outline differences, the position of the anteriormost pore conulus (LM 2) shifts in dorso-ventral direction consistently in both valves and sexes. In females, also the position of the dorsal dip point of the posterior curvature (LM 6) varies in dorso-ventral direction. Shape variation along RW 2 is in females similar as for RW 1 but with a different combination of traits: negative scores correspond to elongate valves with inflated posterior and slightly shifted LM 2 and LM 6 in dorso-ventral direction. In male *Cytheridella*, only left valves show weak biogeographic differentiation along RW 2, representing

shape differences from elongate-elliptical to slightly asymmetrical with higher dorsal margin and the dorsal dip point of the posterior curvature (LM 6) shifted towards posterior. The only differentiation along RW 3 is for male right valves, corresponding mostly to shell elongation and a little to the relative positions of pore conuli.

The PLS analyses show clear relationships between morphological and environmental variables, with similar results across both valves and sexes. The first PLS axis explains between 68.7% and 77.9% of the total variation, whereas values are

consistently higher for females (LV: 77.5%; RV: 77.9%) than for males (LV: 68.7%; RV: 71.5%). In all four analyses, Brazilian specimens are widely separated from Floridian/Mexican ones, corresponding to a clear differentiation along both environmental and morphological scores. Left valves of females and left and right valves of males of Brazilian specimens exhibit negative scores on both PLS axes corresponding to shape and environmental variables. Females display inverse distributions for Brazilian and Mexican specimens. Floridian and Mexican groups overlap little but consistently in all analyses

while the specimens of Florida tend to have smaller variation ranges than Mexican groups (Fig 5).

The loadings for morphological variables yield constantly high values for RW 1; RW 2 is important to all analyses except male right valves; RW 3 contributes to variation in male valves but hardly in females. Higher warps were not considered because of their minor influence on shape variation or the lack of biogeographic separation. See Table 1 for a summary of the results.

Following warps fulfil the selection criteria defined in the methods section for consideration in the multiple regressions: RW 1 for all four datasets; RW 2 for female right and left valves and male left valves; RW 3 for male right and left valves. Hence, nine regression analyses were carried out. Shapiro-Wilks tests of model residuals indicate normality for analyses of males but not females (Table 2). Inspection of Q-Q-plots yielded, however, that in all models the majority of cases match the expected

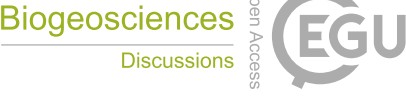



distributions, which is why the models for females are still considered significant (see Appendix S2). Seven out of nine models are significant ($P < 0.05$); the models for RW 2 and RW 3 for male left valves are not ($P = 0.08$ and $P = 0.07$, respectively). Only a limited set of predictor variables is retained out of the originally 15 variables in each model. Seven parameters do not contribute to any models: $Na^+$, $Ca^{2+}$, $Mg^{2+}$, $HCO_3^-$, and conductivity, mean annual temperature and precipitation seasonality.

Of the remaining factors, temperature seasonality is one of the most important predictors in almost all models, accounting for at least 28.7% in all models with RW 1. Temperature seasonality is highest in Florida, closely followed by Brazil, and considerably less in Mexico, reflecting the distinction between Mexican and Floridian/Brazilian populations along RW 1. Similarly, annual precipitation and the anions $Cl^-$ and $SO_4^{2-}$ contribute significantly to many models, corresponding to differences in the hydrological regimes. Less explanation power provides pH, $K^+$, water temperature, and mean diurnal

temperature range. It is noteworthy that anions, represented by $Cl^-$ and $SO_4^{2-}$ are obviously much more important than cations.

## 4 Discussion

Variation in temperature seasonality, annual precipitation and anions ($Cl^-$, $SO_4^{2-}$) explain a large portion of shape variation in *Cytheridella*, which is mostly related to relative carapace length and outline shape. Narrow elongate shapes, such as those occurring in Mexico, correspond to relatively low seasonality and precipitation but high anion concentrations. Opposite

conditions favor the formation of short, asymmetrical valves typical for specimens from Florida and Brazil. Secondary shape variations differentiating between elongated valves with slightly wider posterior and short, symmetrical valves (i.e., RW2) are attributed to higher and lower annual precipitation, respectively.

The link between shape variation and environmental conditions is a well-studied branch of ostracodology, but studies have yielded quite contrasting results. Frequently identified ecological factors are salinity (Yin et al., 1999; Yin et al., 2001; Grossi

et al., 2017) and hydrochemical regime, mirrored by $Mg^{2+}$, $Ca^{2+}$ and $K^+$ contents (Ramos et al., 2017) or alkalinity and sulphate, respectively (van der Meeren et al., 2010). Morphological response to the same environmental factor may even differ between environments (e.g., Yin et al., 1999), complicating straightforward explanation models.

### 4.1 Potential environmental drivers of valve shape variation

The geographical range of *Cytheridella* coincides with the Neotropical region which spans a wide latitudinal range from ~30°N

to ~30°S. This range involves a latitudinal decline in mean annual temperature, which mainly corresponds to differences in annual minimum temperature (Lewis, 1996). Both Florida and southern Brazil are characterized by higher annual temperature gradients compared to Mexico. Annual minimum and maximum temperatures in Florida and S-Brazil range between 16°C and 3°C and 10°C and 30°C, respectively. Minimum and maximum temperatures vary between 19°C and 33°C (Climate-Data.org, 2017).

Temperature has a direct effect on other environmental parameters such as salinity and oxygenation of the water. Water temperature is one of the most important variables affecting metabolism, oxygen consumption, growth, molting and survival





of crustaceans (Le Moullac and Haffner, 2000 and references therein). Increases in temperature can result in significantly shortened intermolt periods, higher molting rates (Mezquita et al., 1999; Brylawski and Miller, 2006), increased growth increments (Martens, 1985; Iguchi and Ikeda, 2004) and reduction in maturation time (Pöckl, 1992). We expect that higher temperature seasonality induced prolonged molt cycles in populations of *C. ilosvayi* by extending intermolt periods during

colder seasons. How these changes affected valve shape is, however, not understood at present.

Precipitation causes declines in nutrients and promotes physical disturbance of the water column (Figueredo and Giani, 2009). Moreover, changes in precipitation directly influence hydrochemical composition, input of sediments, organic components and contaminants and lake level (Mortsch and Quinn, 1996; Whitehead et al., 2009). Indirect influence represent, e.g., the control on aquatic plants, which represent important (micro)habitats and/or food sources (Lacoul and Freedman, 2006). The

annual cycle of precipitation over most of South America is monsoon-like with great contrasts between winter and summer (Grimm et al., 2007). In the region the Brazilian samples derive from the peak rainy season is the austral winter, a mid-latitude regime, where the rainfall is due to frontal penetration associated with migratory extratropical cyclones (Grimm et al., 1998). The amount of rainfall in Yucatan is associated with seasonal migration of the Intertropical Convergence Zone and less by spatially oriented tropical convective activity (e.g., Hodell et al., 2008). Florida, in particular Southern Florida, where most of

the sampled areas derive from, receives maximum precipitation during northern hemisphere summer from convectional and tropical storms (Schmidt et al., 2001). The annual precipitation amounts for the sampled areas are with 1396–1492 mm per year in Brazil higher on average than in Florida and Yucatan, with 1185–1430 mm and 1125–1359 mm, respectively. Since the annual amounts of the regions are very similar it might be more plausible that precipitation seasonality has an influence on carapace shape of *Cytheridella* through seasonally restricted nutrient inputs or changes of the hydrochemistry. Annual

precipitation should be therefore considered with caution since it is difficult to deduce a causal relationship with carapace shape.

Ionic composition of the host water is vital for calcification and growth rates of ostracods (Mezquita et al., 1999). The relationship between hydrochemistry and phenotypic variability is poorly understood, however. A latest study of Kim et al. (Kim et al., 2015) shows that increased levels of pH account for decreased carapace growth rates, i.e., prolonged intermolt

periods, and smaller carapaces. Carapace shape changes have been moreover associated with changes in $Ca^{2+}$, $Mg^{2+}$ and pH (Ramos et al., 2017). Our analyses, however, revealed correlations neither with ions related to formation of carbonate, such as $HCO_3^-$, $Ca^{2+}$, and $Mg^{2+}$, nor with pH. Only chloride and sulfate contents significantly explain carapace shape variation.

Natural sources of $Cl^-$ in freshwater derive from marine sprays transferring NaCl into the atmosphere and are transported as aerosol by wind, and washed out by precipitation and the weathering of rocks. Additionally large amounts of chloride are

derived anthropogenically from farming and waste water production (Müller and Gächter, 2012). Sulfate can derive through runoff from mining and agricultural areas, mobilization from pyrite deposits by oxygen intrusion during desiccation and weathering of rocks containing Sulphur (Holmer and Storkholm, 2001; Lamers et al., 2002). Sulfate contents in groundwaters and surface waters result from dissolution of gypsum and anhydrite occurrences (Perry et al., 2002) and mixing with seawater (Sacks et al., 1995). In Yucatan, a gypsum-rich stratigraphic unit occurs providing a solution-enhanced subsurface drainage





pathway for a broad region extending along the eastern coast and from east to west in the southern part (Perry et al., 2009). The chloride content of groundwater is the result of mixing with seawater (Mondal et al., 2010). Additionally, a Cl⁻ gradient extends from southeast to northwest providing generally higher chloride contents (Perry et al., 2009). Concentration gradients of $SO_4^{2-}$ and Cl⁻ in Florida occur from inland to coastal areas as well as with depth (Sacks et al., 1995), explaining the relatively

higher amount of chloride and sulfate. The comparably low values for south Brazilian sampling locations is not surprising given that such coastal water bodies are often fed by groundwater (Santos et al., 2008) that are dominated by bicarbonate waters and low chloride and sulfate contents (Gianesella-Galvão and Arcifa, 1988; Viero et al., 2009). The detected relationship between morphotypes and chloride and sulfate contents, respectively, could thus mirror the hydrochemical compositions resulting from different hydrogeological conditions of the regions.

(van der Meeren et al., 2010) found ostracod valve shape variability to be significantly correlated with the ratio between alkalinity and sulfate. As the ratio was inversely related to solute concentration, the authors hypothesized that carapace shape may be linked to changes in the lake water balance or relative climatic moisture, or changes in the sources of solutes delivered to the environment. Varying anionic composition has also been considered to affect osmoregulation and calcification (Mezquita et al., 1999). As hyperosmotic organisms, freshwater ostracods are obliged to pump ions inwards (mainly Na⁺ and Cl⁻) and

water outwards to maintain a stable internal ionic concentration higher than that of the ambient water (Weihrauch et al., 2004). Chloride is obtained from the environment through a $HCO_3^-$/Cl⁻ antiport pump. The organism needs to precipitate calcite but also pump $HCO_3^-$ outwards to maintain the internal Cl⁻ concentration (Mezquita et al., 1999). These authors assumed that even small genetic differences affect varied ecophysiological responses to temperature and water chemistry, which may be a key factor for the explanation of different biogeographical patterns of non-marine ostracods. Especially the trade-off between ionic

regulation and calcification is considered to play a key role in ostracod speciation (Mezquita et al., 1999).

### 4.2 Genetic diversity or ecophenotypic plasticity?

Phenotypic variation in ostracods is considered to reflect either genotypic or ecophenotypic variability or a combination of both (Martens et al., 1998; Yin et al., 1999; Anadón et al., 2002; Frenzel and Boomer, 2005; Boomer et al., 2017; Grossi et al., 2017). A recent study on valve outline variability of a non-marine ostracod demonstrated that differences in carapace shape

do not correspond to genetic clades (Koenders et al., 2016). However, caution is advised when comparing patterns among species, since different species react differently and have different potentials for ecophenotypic variation (Anadón et al., 2002; Frenzel and Boomer, 2005). Nonetheless, the here documented morphological differences in carapace shape among populations from Mexico, Florida and Brazil, which are consistent to patterns shown for appendages (compare Wrozyna et al., under review), may argue for genetic differentiation. This hypothesis is supported by the fact that soft parts are assumed to be

more conservative to environmental influences (Park et al., 2002; Park and Ricketts, 2003). Similarly to (Koenders et al., 2016), we therefore assume that the observed morphological cluster correspond to cryptic species. Wrozyna et al. (submitted) tentatively ascribes the currently widespread distribution of the "*Cytheridella ilosvayi* species group" to passive dispersal via birds. Local populations are considered to have adapted to prevailing environmental conditions, followed by morphological





and, eventually, genetic divergence. This highly speculative scenario needs to be tested using molecular data, which are not available at present however.

The relationship between genotype and environment might differ among species, geographical regions and through time (see, e.g., Sanchez-Gonzalez et al., 2004; Koenders et al., 2016). Concluding from our statistical results it is clearly implied that

morphological disparity of *Cytheridella* is controlled by environmental factors. However, the distribution and the variation range of regional clusters reveal some opposing implications. For instance, Florida and Brazil show a clear differentiation on RW2 what could correspond partially to the different annual precipitation amounts. Following this, Mexico, should occur at most positive scores on the RW 2 axes due to lowest annual precipitation amounts (Table 1). But, Mexican specimens are widely distributed on RW 2 with a clear gap differentiating two groups. The smaller segregated group comprises specimens

with similar (shortened) valve shapes as the Floridian group but co-occur with the elongated morphotypes in one lake (i.e., Punta Laguna) (Wrozyna et al., under review). Consequentially, the two morphotypes indicate opposite magnitudes of environmental factors. Regulation of valve shape variability exclusively by environmental parameters is, thus, rather improbable.

Since morphological variability of the appendages reflects, the pattern revealed by hard parts it is assumed that regional

morphotypes of *Cytheridella* represent (cryptic) morphospecies. Wrozyna et al. (submitted) provide a hypothesis that widespread distribution of *Cytheridella* may have originated through, e.g., passive dispersal and how local populations may have adapted to local environmental conditions accompanied by successive morphological divergence and eventually speciation.

A higher morphological similarity between Florida and Brazil can be explained by interconnection of these regions through,

e.g., avian dispersal. Ecological conditions in Mexico (e.g., higher or more variable salinity) could have been promoted the evolution of new species (with more elongated outlines) which have occupied all available niches preventing colonization through other species. This model allows to explain the co-occurrence of two different morphotypes (species) in one lake. It cannot be ruled out that temperature seasonality and chloride and sulfate concentrations have contributed to these processes.

## 5 Conclusion

The comparison of our results and a large number of previous studies witnesses the difficile nature of ecophenotypic response to varying climatic and ecological conditions in freshwater ostracods. Shape variation in *Cytheridella*, mostly related to relative carapace length and outline shape, is mainly explained by temperature seasonality, annual precipitation and chloride and sulfate compositions. Increased temperature seasonality, characteristic for Florida and south Brazil, are considered to account for slower growth rates during colder months and may have triggered the development of shortened valves with well-developed

brood pouches. Differences in chloride and sulfate concentrations, which are also related to fluctuations in precipitation, might have affected valve development via controlling osmoregulation and carapace calcification. These explanation models are,



however, tentative as physiological studies on the influence of changing ecological conditions in non-marine ostracods are still scanty.

Temperature *per se*, salinity (expressed as electrical conductivity) and pH have surprisingly little or no effect on shape variation in *C. ilosvayi*, although these factors have been discussed as important drivers of ostracod ecophenotypy, variably affecting

size, ornamentation and shape. The discrepancies in explanation models suggest that environmental predictors for valve shape are not consistent across non-marine ostracods. The nature of the phenotype–environment relationship likely depends on the choice of the model taxon and ecosystem. On a larger scale, this lack of a general pattern complicates reconstruction of paleoenvironments based on ecophenotypic variation.

Regional differences of climatological and hydrogeological conditions could explain a connection between chloride and sulfate

concentrations with different carapace shapes. Thus, the relationship(s) between climate, hydrochemistry, and carapace shape are not straightforward as anticipated. The variables could have also contributed to the evolvement of new species with different carapace shapes.

**Data availability**

All relevant data are presented within the manuscript or in supplementary material.

**Supplement link**

**Author contribution**

C. Wrozyna, J. Meyer and W.E. Piller carried out sampling of modern *Cytheridella* populations. C. Wrozyna and J. Meyer prepared ostracod material for morphometric analyses. T. A. Neubauer performed statistical analyses. Preparation of the manuscript was done by C. Wrozyna and T. A. Neubauer and, M.I. F. Ramos contributed to the discussion of the results and provided taxonomic advice.


**Competing interests**

The authors declare that they have no conflict of interest.

**Acknowledgements**

We are thankful for Norma L. Würdig (Universidade Federal do Rio Grande do Sul and personnel at CECLIMAR in Tramandai (Brazil) for offering facilities.

We also want to thank Thomas Wagner, University Graz, for helpful comments to the hydrological and hydrogeological discussion. Financial support was provided by the Austrian Science Fund (grant number P26554). T.A.N. was supported by a Just'us postdoctoral fellowship granted by the University of Giessen.



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





**Table 1. Results of multiple regressions. Bio2 – Mean diurnal temperature range (mean of monthly (max – min temperature)), Bio4 – Temperature seasonality (standard deviation * 100), Bio12 – Annual precipitation.**

| *Females LV* | *RWA* | | *PLS* | *Multiple Regressions* | | | |
|---|---|---|---|---|---|---|---|
| | % explained | Reification | Loadings axis 1 (77.55%) | $R^2_{adj}$ | P | Shapiro-Wilks test (P) | Predictor variables |
| RW1 | 33.22 | | -0.5637 | 0.632 | <0.001 | 0.0200 | BIO4 (46.2%), $SO_4$ (19.3%), pH (13.0%), $Cl^-$, Temp., BIO12 |
| RW2 | 19.00 | | 0.4798 | 0.510 | <0.001 | 0.0063 | BIO12 (43.9%), $Cl^-$ (32.9%), K (11.7%), $SO_4$, BIO4 |
| RW3 | 10.72 | | -0.1850 | *n/a* | | | |
| | | | | | | | |
| | | | | | | | |
| | | | | | | | |
| *Females RV* | *RWA* | | *PLS* | *Multiple Regressions* | | | |
| | % explained | Reification | Loadings axis (77.95%) | $R^2_{adj}$ | P | Shapiro-Wilks test (P) | Predictor variables |
| RW1 | 40.39 | | -0.3892 | 0.593 | <0.001 | 0.0079 | BIO4 (44.5%), $Cl^-$ (20.4%), $SO_4$ (18.1%), Temp., K, BIO12 |
| RW2 | 14.89 | | -0.2373 | 0.257 | <0.001 | 0.0040 | BIO12 (48.4%), $SO_4$ (23.5%), $Cl^-$ (20.1%), BIO4 |
| RW3 | 10.50 | | 0.1003 | *n/a* | | | |
| | | | | | | | |
| | | | | | | | |
| | | | | | | | |
| *Males LV* | *RWA* | | *PLS* | *Multiple Regressions* | | | |





|  | % explained | Reification | Loadings axis (67.74%) | $R^2_{adj}$ | P | Shapiro-Wilks test (P) | Predictor variables |
|---|---|---|---|---|---|---|---|
| RW1 | 24.14 |  | -0.4063 | 0.638 | <0.001 | 0.3948 | BIO4 (37.3%), Cl⁻ (34.5%), BIO12 (15.0%), SO$_4$ (10.8%), pH |
| RW2 | 22.25 |  | 0.2359 | 0.058 | 0.082 | 0.4130 | (not significant) |
| RW3 | 13.92 |  | 0.2792 | 0.042 | 0.074 | 0.0017 | (not significant) |
|  |  |  |  |  |  |  |  |
|  |  |  |  |  |  |  |  |
|  |  |  |  |  |  |  |  |
| *Males RV* | *RWA* |  | *PLS* | *Multiple Regressions* |  |  |  |
|  | % explained | Reification | Loadings axis (71.55%) | $R^2_{adj}$ | P | Shapiro-Wilks test (P) | Predictor variables |
| RW1 | 29.88 |  | -0.3798 | 0.318 | 0.001 | 0.2221 | BIO12 (31.8%), BIO4 (28.7%), Cl⁻ (15.8%), pH (12.5%), K (11.2%) |
| RW2 | 12.81 |  | 0.0157 | *n/a* |  |  |  |
| RW3 | 11.98 |  | 0.5502 | 0.286 | <0.001 | 0.2978 | BIO4 (57.2%), BIO2 (42.8%) |





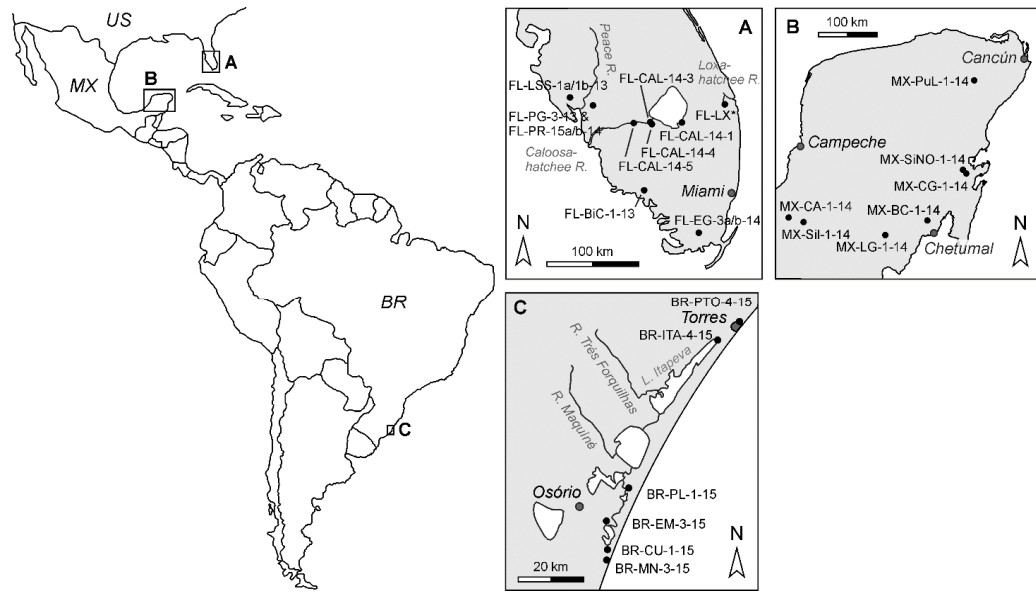

**Figure 1: Geographic overview of the sampled populations (modified from Wrozyna et al. 2016). The label FL-LX\* in map A**
**comprises samples FL-LX-1-14 to FL-LX-6-14. For details, see Table S1.**




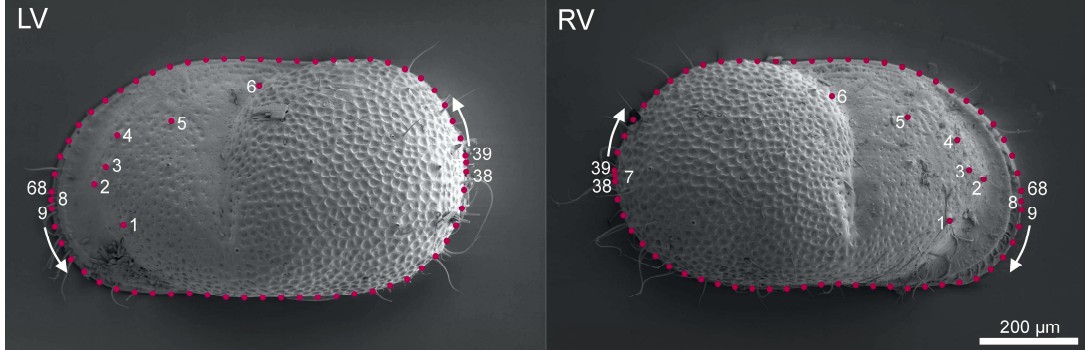

**Figure 2: Landmark configuration on left and right valves of *Cytheridella*. Displayed are female valves. Modified after Wrozyna et al., (2016).**



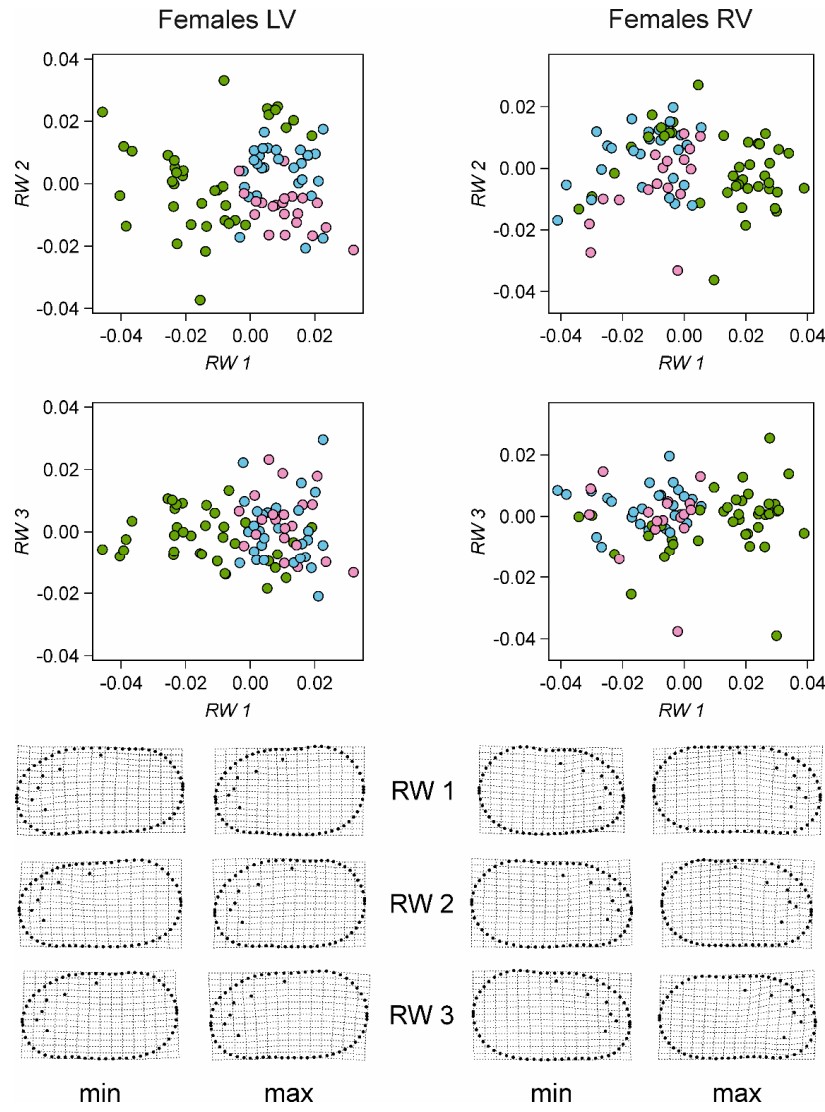

**Figure 3: Relative Warps Analyses of left (LV) and right (RV) valves of females of the first three warps (RW 1 – 3) and the associated thin-splate splines at minimum and maximum scores. Colors refer to the different regions: blue – Florida, green – Mexico, pink – Brazil.**



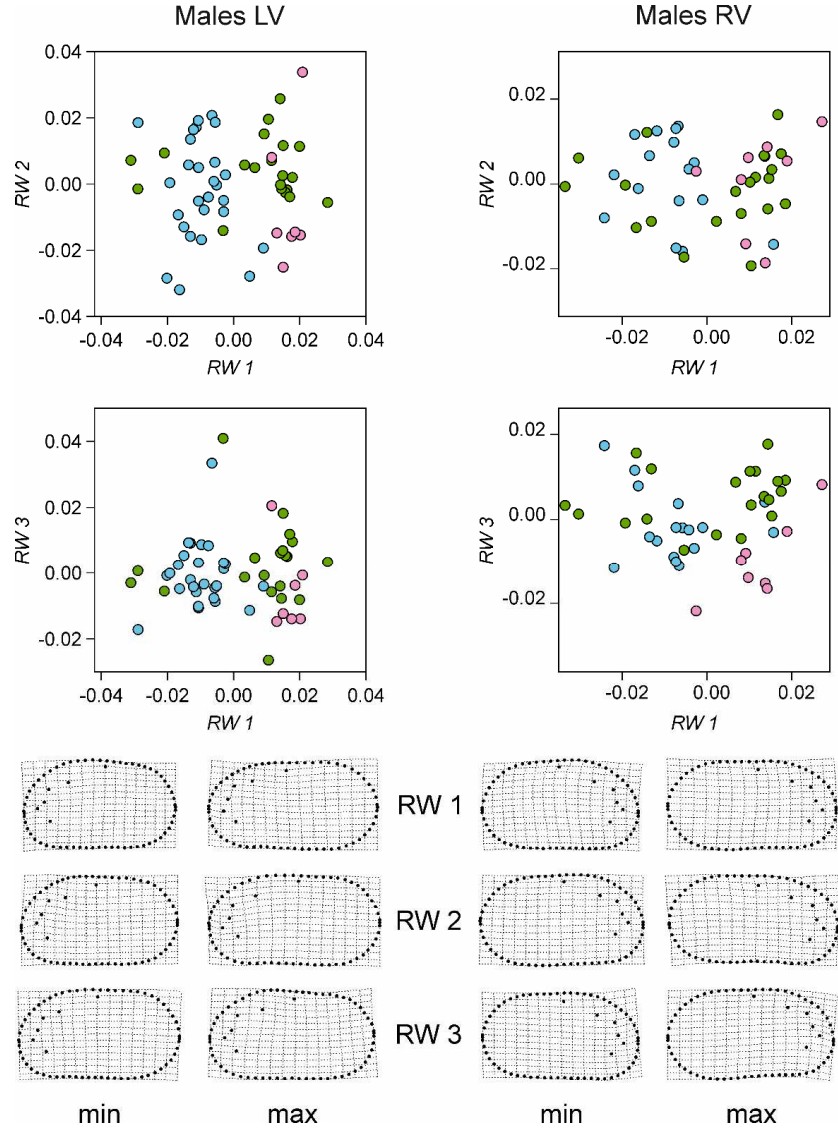

**Figure 4: Relative Warps Analyses of left (LV) and right (RV) valves of males of the first three warps (RW 1 – 3) and the associated thin-splate splines at minimum and maximum scores. Colors refer to the different regions: blue – Florida, green – Mexico, pink – Brazil.**

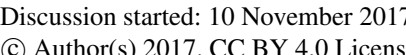
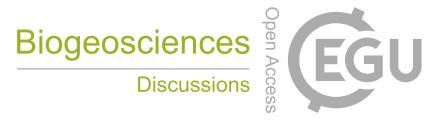


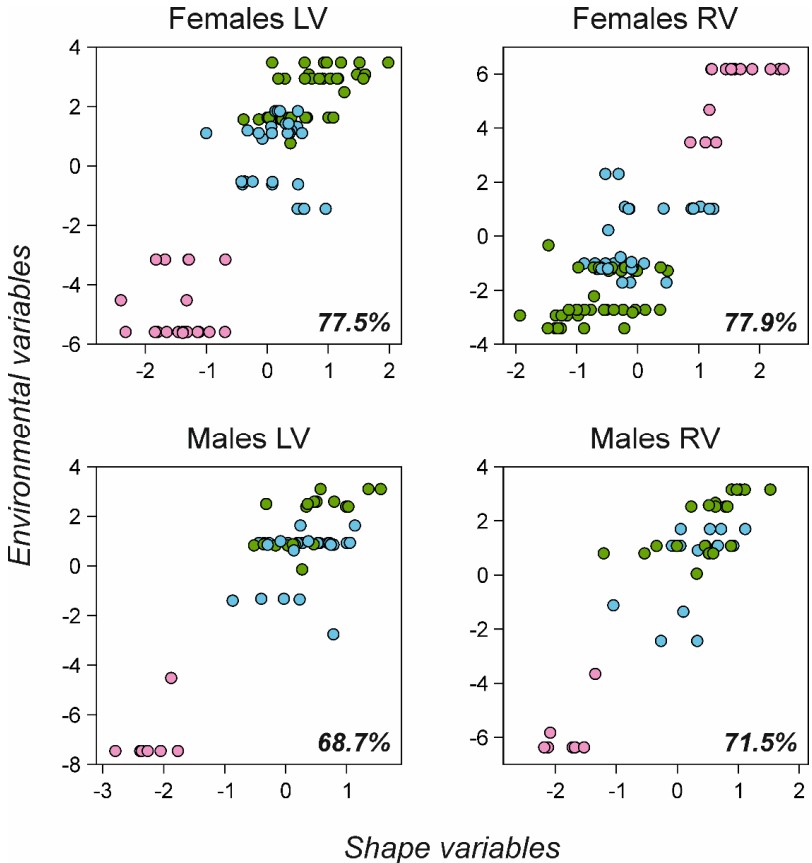

**Figure 5: Partial least-squares analysis of carapace shape of females and males and environmental variables of left and right valves (LV and RV, respectively). Colors refer to the different regions: blue – Florida, green – Mexico, pink – Brazil.**

