# Peer review of "Significance of climate and hydrochemistry on shape variation – a case study on Neotropical cytheroidean Ostracoda"

_Biogeosciences, 2017_

## Referee Comment (RC1) · Anonymous Referee #1 · 24 Nov 2017

General comments

This paper reports the morphological difference of ostracod valves among three geographic regions and its relation to climatic and hydrochemical factors. The research theme is interesting and appealing, but there are some major concerns in analysis and discussion. Further analyses and discussion may improve the paper; therefore, major revision is recommended.

The authors analyzed data by relative warp analysis, PLS analysis, and regression analysis. One of my major concerns is why the authors did not conduct MANOVA/MANCOVA or Procrustes ANOVA, but conducted regression analysis on

each relative warps. The former is appropriate and usually applied for understanding effects of covariates on shape deformation, and is implemented for example in tpsRegr (software by Rohlf FJ, available at http://life.bio.sunysb.edu/morph) and geomorph package in R (Adams & Otárola-Castillo 2013). Moreover, if geographic distance or genetic distance among three geographic regions can be incorporated as phylogenetic information (geomorph implements this analysis), the effects of climatic and hydrochemical factors will become clear. Each RWs as response variables might not reflect shape deformation that covariates with climatic and hydrochemical factors.

In addition, further inspection on the result of PLS analysis is recommended. The authors emphasize the similarity between Florida and Brazil based on the plot of RW1 vs RW2, but Fig. 5 indicated the similarity between Florida and Mexico. Graphical presentation of shape deformation indicated by PLS singular axis of shape variables should be added for further discussion. Loadings of environmental variables in PLS singular axis will be helpful for understanding the effects of climatic and hydrochemical factors. The authors reported first PLS singular axes, but second PLS singular axes might also indicate relationship between shape and environmental factors.

Discussion in the paper descriptively indicates that there are morphological differences that possibly relates to environmental variables, but there are little discussions on why shape of ostracod valves differ depending on environments (e.g. are there any functional meanings? or merely due to physicochemical consequences?) and how ostracods respond to environmental changes. It is impossible to achieve definitive conclusions in the paper, but at least proposing some hypotheses is required.

Literature cited Adams DC, Otárola-Castillo E (2013) geomorph: an R package for the collection and analysis of geometric morphometric shape data. Methods Ecol Evol 4: 393−399.

Specific comments

Page 3 Line 1: The authors focus on "the causes for the regional differences", so how

the factors cause the morphological differences should be more discussed in Discussion.

Line 8: For identifying "the morphological characteristics and environmental variables that contribute most to the relationship", my recommendation is to inspect further the result of PLS analysis and to apply MANOVA/MANCOVA or Procrustes ANOVA, rather than regression analyses conducted in the study.

Line 20: Add explanation more about "water sampling".

Page 5 Lines 9–10: Even if selecting variables that contain high loading values in the PLS analysis, separately applying regression analyses might diffuse the environmental effects on shape.

Lines 11–12: The authors concern multicollinearity. One idea is to conduct principle component analysis of environmental variables, and use PC scores as explanatory variables.

Page 6 Line 18: Show the result of statistical testing of the "clear relationships" in the PLS analyses. Statistical testing is possible by using resampling technique, such as permutation test.

Lines 27–28: Table 1 does not show that higher warps (the authors mean RW4, RW5, etc.?) have minor influence on shape variation. In this paragraph, do the authors show the results of relative warp analysis?

Page 8 Lines 5 & 23: The authors only noted that the reasons why shape differences relate to environmental factors are not or poorly understood. However, at least proposing some hypotheses is required.

Page 9 Lines 22–29: Incorporating phylogenetic information in the analysis may be helpful.

Page 10 Lines 6 & 19: The authors emphasize the similarity between Florida and

Brazil, but the results of PLS analyses seem to indicate the similarity between Florida and Mexico.

Technical corrections

Page 2 Line 1: "individuals" instead of "species" may be better.

Line 9: "extant and extinct" instead of "recent and fossil" may be better.

Page 3 Line 2: "is the knowledge of . . . restricted" should be "the knowledge of . . . is restricted".

Page 4 Lines 17–18: The types mean Bookstein's (1991) types?

Page 5 Line 28: I could not understand which specimens are from Punta Laguna in northern Yucatan. Please show them in Figures 3 and 4.

Line 30: What are "both clusters"?

Page 6 Lines 10–11: The direction of the valve (i.e. antero-posterior axis and dorso-ventral axis) is suggested to be clarified in Fig. 2. Clarify what "consistently" means here.

Page 7 Line 1: "significant" is confusing because it seems to mean "statistically significant". Here, for example, "valid" is better.

Lines 27–28: The sentence is a little complicated. Suggestion: Annual minimum and maximum temperatures in Florida are 16°C and 3°C, and those in southern Brazil are 30°C and 10°C, respectively.

Page 9 Line 5: "comparatively" or "relatively" instead of "comparably"? And low values of what?

Table 1 The table is complicated and difficult to understand. Captions and/or footnotes should be added for making it stand-alone.

[Figure]

---

## Referee Comment (RC2) · Anonymous Referee #2 · 4 Dec 2017

Presented data and descriptions in this manuscript include interesting and important information. The research focus on the influence of the environmental conditions on the geographical-scale morphological variation of ostracod valves was interesting. However, the descriptions of discussion section were too descriptive, and telling only that morphological differences were found among the remotely distributed sites, and those differences seemed to be related to environmental gradients. This is the weakest point of this manuscript as an original article. I feel that the authors should discuss about the mechanisms behind the apparent relationship between shape of ostracod valves and environmental parameters. It would be better if the authors could show some hypothetical interpretation from their observed results. I think deeper discussions on this

point contribute to improve the scientific significance of this manuscript.

---

## Referee Comment (RC3) · Anonymous Referee #3 · 7 Dec 2017

Authors studied about the relationships between geographical variation and some environmental variables. This manuscript is relatively well written, and these results are thought-provoking. However, I hesitate to accept this MS in this high impact journal.

The authors observed meaning patterns between Environmental variables and shape of Ostracods valves. However, I think it is difficult to find definite causal relationships between them from these results. Sure, authors cite some studies that some environmental variables such as temperature crustaceans' molting, maturation and growth. However, to explain the underlining mechanisms that drive shape variation, it is logically invalid. If there were some complementary studies that show relationship between

some variables (e.g. temperature, Cl-, SO42-) and valve shape, authors should add.

If any, author should describe the effects of change in morphological characteristics on ecological significance. Such as effects on production and behavior.

[P2] L11- There have been many studies that environmental variables and morphology. Authors have to show the importance of focusing Ostracods. Authors is just writing "Ostracods represent a model group for the study of ecophenotypical variation. . .". The reasons why Ostracods are model organisms should be described. And the ecological value of Ostracods should be explained.

L12: calcitic → calcific?

[P3] L13- You should describe the brief life cycle of Cytheridella, such as dispersal patterns,life-span, reproduction.

L22-23: Please write about storing condition. In a freezer?

[P9] L22-L2(P10): Authors guess the relationships between genetic differences and carapace shape at the discussion part, lengthy. It is not good to describe lengthy about what is not contained your data. Please delete this speculative sentences.

Fig. 3-5 The symbol of data from "Punta Laguna" should be changed.

---

## Author Comment (AC1) · 25 Jan 2018

We thank the referee for his/her constructive comments on the open discussion paper. Although he/she thinks it is an interesting study, critics is formulated on the methodology of the statistical analyses and the discussion of the relationship between climate and shape changes.

In particular, reviewer's major concern is that our analyses focused on PLS analyses and not (as usual) MANOVA/MANCOVA or Procrustes ANOVA.

- The suggested types of analyses (MANOVA/MANCOVA as well as the inclusion of

phylogenetical data) are used to discriminate groups since they maximize the variance between groups. We know already from previous studies (which are cited in the manuscript) that the groups are different. The present study aims to investigate potential relationships between environmental factors and phenotypes on a regional scale. This requires different kind of analyses. Maybe this was overlooked by the referee. Nonetheless, we will carefully check our introduction in order to emphasize this important difference to conventional studies on morphological variability.

Referee 1 also criticizes that we performed regression analyses on each relative warp.

- This approach enables direct evaluation which environmental factors are associated with which kind of shape variation. We will add this information into the manuscript in order to highlight the strength of this approach.

"In addition, further inspection on the result of PLS analysis is recommended. The authors emphasize the similarity between Florida and Brazil based on the plot of RW1 vs RW2, but Fig. 5 indicated the similarity between Florida and Mexico. Graphical presentation of shape deformation indicated by PLS singular axis of shape variables should be added for further discussion. Loadings of environmental variables in PLS singular axis will be helpful for understanding the effects of climatic and hydrochemical factors. The authors reported first PLS singular axes, but second PLS singular axes might also indicate relationship between shape and environmental factors."

- We will include a sentence about the results of the second PLS axis in the manuscript. However, we suggest providing the detailed results and graphics in the supplements because of the minor relationship between morphology and environment. - We will add the loading values into the supplementary material. We concede that the selection criteria for the warps were not clearly formulated in the manuscript. Therefore, we decided to consider relative warps with PLS loading values higher than the mean loading value (based on absolute values). Accordingly, we added the regression analysis for RW3 of females LV. The values will be changed in the table. These changes do not

affect the interpretation in any way.

"Discussion in the paper descriptively indicates that there are morphological differences that possibly relates to environmental variables, but there are little discussions on why shape of ostracod valves differ depending on environments (e.g. are there any functional meanings? or merely due to physicochemical consequences?) and how ostracods respond to environmental changes. It is impossible to achieve definitive conclusions in the paper, but at least proposing some hypotheses is required."

- Showing that the shape of ostracod valves can vary with respect to differences in environmental conditions is a key finding to understand how these organisms react to external influences. However, the knowledge about physiological processes, including molting and calcification of ostracod valves in relation to environmental parameters, is quite scarce. A further limitation that hinders a straightforward interpretation is the inconsistent results of different approaches of mesocosm experiments. Nonetheless, we will extend our discussion according to the referee's comments, including the consideration if the shape changes might be the result of functional advantages or if they are the result of physiological processes. Since we took our samples from many different habitats, a functional adaptation of carapace shape changes would cause a pattern of similar habitats associated with similar morphotypes independently of the region. Yet, each region is characterized by its own morphotype. This discrepancy rather contradicts the idea of functional morphology. It is therefore more likely that the observed shape changes are caused by physiological processes. However, we would like to avoid discussing hypotheses that cannot be tested in the frame of this paper and prefer a rather tentative way of discussion.

Specific comments Page 3 Line 1: The authors focus on "the causes for the regional differences", so how the factors cause the morphological differences should be more discussed in Discussion. - We carefully revise our discussion and, as mentioned above, try to deepen discussion how the factors cause morphological differences.

Line 8: For identifying "the morphological characteristics and environmental variables that contribute most to the relationship", my recommendation is to inspect further the result of PLS analysis and to apply MANOVA/MANCOVA or Procrustes ANOVA, rather than regression analyses conducted in the study.

- As we discussed already above, MANOVA/MANCOVA or Procrustes ANOVA would be suitable analyses if we were interested in group differences. However, the present study focuses on individual relationships between morphological and environmental parameters, and regression analysis are a convenient tool to investigate those.

Line 20: Add explanation more about "water sampling".

- We will add more detailed information.

Page 5 Lines 9–10: Even if selecting variables that contain high loading values in the PLS analysis, separately applying regression analyses might diffuse the environmental effects on shape.

- The PLS analysis is an overall indication that there is a relationship but the regression analyses allow a more detailed perspective. High PLS loading values indicate a strong relationship between shape and environment. Therefore, we consider it indeed necessary to go into more detail about the relationships with specific warps (and thus morphological traits).

Lines 11–12: The authors concern multicollinearity. One idea is to conduct principle component analysis of environmental variables, and use PC scores as explanatory variables.

- Multicollinearity is a natural phenomenon in datasets with many variables, particularly environmental parameters use to be correlated to some extent. As stated clearly in the respective paragraph, we apply stepwise selection of variables based on the variance inflation factor. This procedure prevents misinterpretation from correlated variables.

Page 6 Line 18: Show the result of statistical testing of the "clear relationships" in the

PLS analyses. Statistical testing is possible by using resampling technique, such as permutation test.

- We will apply permutation tests on the PLS results.

Lines 27–28: Table 1 does not show that higher warps (the authors mean RW4, RW5, etc.?) have minor influence on shape variation. In this paragraph, do the authors show the results of relative warp analysis?

- This paragraph is about the PLS analysis. We will reformulate it to avoid further confusion.

Page 8 Lines 5 & 23: The authors only noted that the reasons why shape differences relate to environmental factors are not or poorly understood. However, at least proposing some hypotheses is required.

- See above.

Page 9 Lines 22–29: Incorporating phylogenetic information in the analysis may be helpful.

- This is an interesting point but this was not the intention of the present study.

Page 10 Lines 6 & 19: The authors emphasize the similarity between Florida and Brazil, but the results of PLS analyses seem to indicate the similarity between Florida and Mexico.

- This contradiction is caused by mixing two different aspects. The emphasis of the similarity between Florida and Brazil refers to the morphologies (given in the Relative Warps Analyses), both regions provide rather shortened carapace outlines compared to Mexican carapaces which are strikingly different. The PLS indeed reflects the relationship between environmental gradients and shape changes. The higher 'similarity' of Florida and Mexico in the PLS is caused by similar environmental factors. For instance, Florida and Mexico cover wider conductivity ranges (205 to 2360 $\mu$S/cm)

whereas Brazil is represented by low conductivities ($\leq 279$ $\mu$S/cm). We will rephrase this to avoid misinterpretation.

Technical corrections will be done according to the referee's comments.

---

## Author Comment (AC2) · 25 Jan 2018

Referee #2 evaluates the manuscript providing interesting and important information. He emphasizes the research focus on the influence of the morphological variability on a geographical scale. His concerns follow referee #1 demanding some hypothetical interpretations about the mechanisms behind the relationship between shape of ostracod valves and environmental parameters.

- We agree with the referee that the manuscript will benefit from an extended discussion. We stated above that straightforward interpretations of these mechanisms are hindered due to lacking and/or inconsistent studies. Moreover, our study is not suited

for testing those hypotheses for which physiological experiments would be required. For these reasons we kept the discussion in a rather reduced form in order to avoid speculative hypotheses that cannot be tested. Nevertheless, as written above, we will extend our discussion.

---

## Author Comment (AC3) · 25 Jan 2018

Also referee #3 characterizes the manuscript positively and appoints the results as thought-provoking. His concerns are similar to the both other referees and refer to missing explanation for the mechanisms that drive shape variation. He/she suggests adding complementary studies that show relationship between environmental variables and valve shape. It is also noted that we should describe the effects of change in morphological characteristics on ecological significance such as effects on production and behavior.

- As described above, we comprehend that from the referee's perspective it is necessary that the mechanisms of the ecologically induced shape changes should be explained. For the aforementioned reasons, these interpretations remain very speculative. Nonetheless, we will check our discussion carefully and try to hypothesize possible mechanisms (see reply to referee #1, functional vs. physiological meaning). - Indeed, the understanding of the ecological significance on, e.g., production and behavior, would be of great importance. Yet, information on ostracod biology is quite fragmentary. It is questionable whether the ecological significance of morphological changes in other, better-studied organism groups is comparable to the present study. However, we will check the literature for appropriate studies.

[P2] L11- There have been many studies that environmental variables and morphology. Authors have to show the importance of focusing Ostracods. Authors is just writing "Ostracods represent a model group for the study of ecophenotypical variation . . .". The reasons why Ostracods are model organisms should be described. And the ecological value of Ostracods should be explained.

- We wanted to avoid a lengthy introduction therefore kept it as short as possible. We will add the explanation why they are considered as model organisms. As far as possible, we will include information about their ecological value.

L12: calcitic ! calcific?

- Calcitic is the correct term for describing materials that are made of calcite (as ostracode valves).

[P3] L13- You should describe the brief life cycle of Cytheridella, such as dispersal patterns, life-span, reproduction.

- In fact, our data set provides the first in-depth information about distribution and environmental parameters (e.g., conductivity, pH, . . .). This is astonishing since Cytheridella is one of the most widespread Neotropical ostracod species. However, information about life cycle, life span and reproduction is not available from the literature

and cannot be deduced from our data. That is why the interpretation of the underlying physiological processes of the shape-environmental relationship is so difficult. L22-23: Please write about storing condition. In a freezer?

- We stored samples in a freezer. Details will be supplied in the revised manuscript version.

[P9] L22-L2(P10): Authors guess the relationships between genetic differences and carapace shape at the discussion part, lengthy. It is not good to describe lengthy about what is not contained your data. Please delete this speculative sentences.

- Unnecessary information will be deleted to shorten the discussion.

Fig. 3-5 The symbol of data from "Punta Laguna" should be changed.

- The symbol will be changed.
* * *

---

## Author Response (AR1)

**Revision of Manuscript bg-2017-390**

**`Impact of climate and hydrochemistry on shape variation – a case study on Neotropical cytheroidean Ostracoda´**

5  As indicated in our reply to the reviewer comments we have revised our manuscript carefully.

**Revisions according to Reviewer #1**

In particular, reviewer's major concern is that our analyses focused on PLS analyses and not (as usual) MANOVA/MANCOVA or Procrustes ANOVA.

> ➢ We have carefully checked our introduction. It is indicated in p2 lines 4-10 that the biogeographical variation is known, but the present study focus on regional differences that have not been investigated, yet. Therefore, we have applied two-block partial least squares analyses that enable to study identify the morphological characteristics and environmental variables that contribute most to the relationship (see p2 line 9-10).

Referee 1 also criticizes that we performed regression analyses on each relative warp.

> ➢ The respective section was rephrased in order to highlight that this approach enables direct evaluation which environmental factors are associated with which kind of shape variation. See p4 line 29 -p5 line 3.

20  "In addition, further inspection on the result of PLS analysis is recommended. The authors emphasize the similarity between Florida and Brazil based on the plot of RW1 vs RW2, but Fig. 5 indicated the similarity between Florida and Mexico. Graphical presentation of shape deformation indicated by PLS singular axis of shape variables should be added for further discussion. Loadings of environmental variables in PLS singular axis will be helpful for understanding the effects of climatic and hydrochemical factors. The authors reported first PLS singular axes, but second PLS singular axes might also indicate

25  relationship between shape and environmental factors."

> ➢ We have included a sentence about the results of the second PLS axis in the manuscript (p 6 line 26-27).
> ➢ As suggested we providing the detailed results in the supplements because of the minor relationship between morphology and environment.
> ➢ Also, we added the loading values into the supplementary material. We formulated the selection criteria for the
30  warps more clearly in the manuscript. Accordingly, we added the regression analysis for RW3 of females LV. The values will be changed in the table.

"Discussion in the paper descriptively indicates that there are morphological differences that possibly relates to environmental variables, but there are little discussions on why shape of ostracod valves differ depending on environments (e.g. are there any

functional meanings? or merely due to physicochemical consequences?) and how ostracods respond to environmental changes. It is impossible to achieve definitive conclusions in the paper, but at least proposing some hypotheses is required."

> ➢ We will extend our discussion according to the referee's comments, including the consideration if the shape changes might be the result of functional advantages or if they are the result of physiological processes. See p 7 line 28 – p 8 line 5 and page 10 line 4 -14.

Page 3 Line 1: The authors focus on "the causes for the regional differences", so how the factors cause the morphological differences should be more discussed in Discussion.

> ➢ See above. We extended the discussion with respect to explain morphological differences.

Line 20: Add explanation more about "water sampling".

> ➢ The information were added.

Page 6 Line 18: Show the result of statistical testing of the "clear relationships" in the PLS analyses. Statistical testing is possible by using resampling technique, such as permutation test.

> ➢ We have applied permutation tests on the PLS results. Results are given at p6 line 20-25.

Lines 27–28: Table 1 does not show that higher warps (the authors mean RW4, RW5, etc.?) have minor influence on shape variation. In this paragraph, do the authors show the results of relative warp analysis?

> ➢ This paragraph is about the PLS analysis. We have reformulated it to avoid further confusion.

Page 8 Lines 5 & 23: The authors only noted that the reasons why shape differences relate to environmental factors are not or poorly understood. However, at least proposing some hypotheses is required.

> ➢ See above. As indicated we have inserted some hypotheses.

Technical corrections were done according to the referee's comments.

**Revisions according to Reviewer #2**

His concerns followed referee #1 demanding some hypothetical interpretations about the mechanisms behind the relationship between shape of ostracod valves and environmental parameters.

> ➢ See above. The discussion was extended and includes some hypothetical interpretations about the mechanisms behind the relationship between shape of ostracod valves and environmental parameters.

**Revisions according to Reviewer #3**

His concerns are similar to the both other referees and refer to missing explanation for the mechanisms that drive shape variation. He/she suggests adding complementary studies that show relationship between environmental variables and valve shape. It is also noted that we should describe the effects of change in morphological characteristics on ecological significance such as effects on production and behavior.

➢ We have checked our discussion carefully and hypothesized possible mechanisms (see reply to referee #1, functional vs. physiological meaning).

➢ Since information on ostracod biology is quite fragmentary we have not found for appropriate studies that could have contribute to the present study.

L22-23: Please write about storing condition. In a freezer?

➢ Information were added.

[P9] L22-L2(P10): Authors guess the relationships between genetic differences and carapace shape at the discussion part, lengthy. It is not good to describe lengthy about what is not contained your data. Please delete this speculative sentences.

➢ Unnecessary information was deleted to shorten the discussion.

Fig. 3-5 The symbol of data from "Punta Laguna" should be changed.

➢ We reconsidered the request of the reviewer and decided for reasons of consistency not to follow him/her. The manuscript focuses on regions not on populations. The population "Punta Laguna" is mentioned since there are two morphotypes which causes an overlap of the Floridian and Mexican morphospace. However, this is sufficiently discussed in our other studies which are cited.

➢

[revised manuscript text omitted]

---

## Author Response (AR2)

Dear Editor,

I am sending you our revised manuscript "Impact of climate and hydrochemistry on shape variation – a case study on Neotropical cytheroidean Ostracoda" and replies to the referee comments.

5   The repeated critics of referee #1 are related to the statistical approach and the request to provide explanations how environment impacts on the carapace shape. We want to emphasize here once more that the statistical approach using PLS analyses was chosen to test for the overall covariation between morphology and environment. Because of the complexity of environment and shape, which the referee is explicitly referring to, we used multiple regressions that allow a more precise estimation of the parameter affecting specific shape traits. It is the combined

10  power of PLS and multiple regressions that help to disentangle the relationships of shape and extrinsic factors.

We cannot follow the referee's comment that explanations for how environmental changes influence shape changes are important for considering acceptance for publication. Despite the lacking physiological explanations (which are simply not possible from out data), our study is the first that characterizes the relationship of environmental variables and associated morphological changes on a large geographical scale. Moreover, we do discuss the results

15  as broadly as possible. We believe that our study provides a useful baseline for further, complementary studies with more specific research objectives such as the physiological processes involved in morphological changes.

Referee #3 seemed to be uncertain about the novelty of the study. In order to clarify that point, we have emphasized in the MS that the dataset covering a supra-regional scale and hydrochemical and climatic data represent a novel contribution to research on ecophenotypy.

20  Our study provides new insights and thought-provoking impulses in terms of species–environment interactions and can be important for interpreting potential future studies. As such we think this manuscript is of interest not only to specialists on ostracods or freshwater ecologists but to a broad readership of geoscientists and biologists in general.

Please find more in-depth replies to the referees' comments attached.

Yours sincerely,
Claudia Wrozyna

**Reply to Anonymous Referee #1 (Report submitted on 13 June 2018)**

5 We thank the referee for his/her constructive comments on the revised manuscript.
His/her major critics refer to the inferences of the PLS analyses and non-reasonable explanations for how the environment impacts shape variation in *Cytheridella*.

10 "I have checked the revised manuscript, but the revised manuscript does not reply or refute to some of my concerns. For example, the places of the specimens from Punta Laguna in the morphospace (Figs. 2 and 3) are not clarified."

➢ In our first revision and reply to the referees comments we have carefully checked all of his/her requests. We obviously overlooked this point. We will prepare a revised figure indicating Punta Laguna.

"1) Because PLS maximizes the correlation of shape variables and environmental variables, I think authors should evaluate the results of PLS carefully. Graphical presentation of shape deformation indicated by PLS singular axis of shape variables should be added. In the first PLS singular axis, there seems to have no excess values of loadings in specific environmental variables. This may be due to the fact that the environment is complex. So simplifying
20 the focus on specific environmental variables may be misleading, despite selection of variables by VIF. Although the similarity between Florida and Brazil is emphasized, the results of PLS analyses seem to indicate the similarity between Florida and Mexico. The discrepancy is not mentioned in the responses."

➢ There seems to be a confusion here. The PLS loading values for the environmental variables are partly quite high and do warrant an interpretation. However, we intentionally did not go into the details for the
25 PLS exactly because of the complexity of environment and shape. In fact, the complete picture presented by the PLS might not be able to show the effect of environmental parameters on specific shape traits. Also, we did not stick to "specific environmental variables" as claimed by the referee; the multiple regressions included several parameters, only we focused on single warps that correspond to specific shape traits. An aim of the paper was to find out which environmental parameters/combination of
30 parameters are responsible for differences in which shape trait(s). This question was answered using multiple regressions and could not be done so only using PLS. Moreover, the software we used for the PLS analyses does not offer the possibility of generating thin-plate splines (but we do not consider this a drawback because the stronger focus in on the regressions).

➢ Concerning the similarities of the three regions, the referee seems to have overlooked that we had already
35 responded in our first reply to the apparent "discrepancy" of the similarities. This contradiction is caused by mixing two different aspects. The emphasis of the similarity between Florida and Brazil refers to the morphologies (given in the Relative Warps Analyses), both regions provide rather shortened carapace outlines compared to Mexican carapaces which are strikingly different. The PLS indeed reflects the

relationship between environmental gradients and shape changes. The higher `similarity´ of Florida and Mexico in the PLS is caused by similar environmental factors. For instance, Florida and Mexico cover wider conductivity ranges (205 to 2360 µS/cm) whereas Brazil is represented by low conductivities (≤ 279 µS/cm). We believe this is clearly stated in the text.

"In this revision, authors added some explanations, but they are not so persuasive or well-structured for emphasizing "impact of climate and hydrochemistry". The authors indicate the environmental difference among
10 regions, but do not provide reasonable explanations for how the environment impacts on shape variation. Because the novelty of this manuscript is the relationship between shape and environment (the geographic variation of shape seems to be another study by authors; Wrozyna et al. 2016, under review), this is important for considering acceptance for publication."

15 ➤ The referee asks for "reasonable" explanations how the environment impacts on shape regions, but it is unclear what kind of discussion he/she wants to see exactly. Since no further comments are provided to explain this, we assume that he/she expects physiological explanations, i.e., environmentally induced processes that lead to shape changes of the carapaces. We fully agree with the referee that this would be highly required especially since it could contribute to understand if the control of environmental
20 parameters on morphology are, e.g., species-specific, related to geographical scale (local populations vs. geographical range of a species or habitat type), interact with other processes and/or parameters, etc. However, this requires complementary physiological studies that are unfortunately unavailable as yet.
➤ Nevertheless, we do provide explanation models in the discussion. The key process that combines causes and consequences is the calcification of new valves during molting. Due to the small size of ostracods
25 and the rapidity of the process (usually hours to few days) field studies are not the appropriate approach. Mesocosm experiments, in turn, allow the observation of specimens under controlled hydrochemical conditions. Previous ostracod-related mesocosm experiments have been studied with respect to restricted number of variables with a clear focus on the effects of salinity changes.
➤ Our dataset contains not only salinity and other principle hydrochemical variables but also climatic data.
30 This is a major difference to all other previous studies. They are dealing either with mesocosm experiments or with (a restricted number) of field populations and hydrochemical data only. Hydrochemical data can vary on very small spatial scales in contrast to climatic data. Thus, so far only local or regionally-restricted relationships have been considered. While we do not want to lower the relevance of these previous studies, our inter-regional approach is the first ever that investigates
35 environmental impacts on the morphology on a (large) geographical scale that coincides with the geographical range of the investigated species and allows inferences about the environment-shape relationships inter-regionally.
➤ In summary, we cannot provide explanations how environment impacts on shape in terms of physiological processes, but we think that our study provides new insights and thought-provoking
40 impulses in terms of species-environment interactions and can be important for interpreting potential

future studies. Nonetheless, we extended the discussion to emphasize the novelty of the study in order to meet the referees´ critics that explanations are not persuasive and well-structured.

➢ In order to follow the referees´ concerns that the title/statement of the manuscript is not justified we change it into: "Significance of climate and hydrochemistry …".

**Reply to Anonymous Referee #3 (Report submitted on 09 May 2018)**

We thank the referee for his/her constructive comments on the revised manuscript.
He/she formulates critics on the missing emphasis on the novelty of the study.

"Authors revised the MS accordance with referee's comments. Especially, to explain the mechanisms that can change the valve shape, authors revised MS based on a lot of previous studies. However, authors also had to emphasize the novelty of study, in this revised version. Authors introduce a lot of previous studies that show relationships between ecophenotypic variation of freshwater ostracods and environmental variables. This is fine but the value of this study become ambiguous."

> The introduction of previous studies is necessary in order to show inconsistencies in the approaches to study ecophenotypical characteristics as well as the conclusions made from the different studies. Furthermore, this provides the motivation for the present manuscript and its novel approach. Nevertheless, we added a sentence at the end of the introduction to explicitly highlight the novelty of the study.

"In addition, true, valve shape seems to be influenced by some environmental variables but the effect of random genetic drift was not excluded. Authors should discuss about it."

> We agree with the reviewer that genetic drift must be considered for the interpretation of morphological variation. Indeed, we had the intention to characterize the genetic divergence of the different morphotypes. However, due to unknown reasons these analyses failed (B. Stelbrink, University of Giessen, personal communication). In order to avoid extensive discussion that cannot be tested by our data, we restricted the discussion to the comment that genetic differentiation might be an explanation for some discrepancies such as the co-occurrence of morphotypes in one locality. Any further discussion would be highly speculative.

Minor comments
· I commented you should introduce life cycle of Cytheridella. Not every reader is specialist about Cytheridella, especially in journals like BGS that accept variety field studies. Current MS is a little hard to understand the significance of this study, if he/she don't know ecological information, such as life span and dispersal ability that affect morphological evolution.

> We totally agree with the reviewer that the manuscript would benefit from such important information. This was already requested by one of the previous reviewers. Until then, information about life cycle, life

span and reproduction were not available for *Cytheridella* in the literature and cannot be deduced from our data. However, a recently published paper of our working group provides now more information on life span, molt cycles and calcification periods of *Cytheridella*. We will add that information in the revised discussion.

• Although paper writing style differ from person to person, I think the word "hypothesize" should not be used in discussion part. Of course, I know this revision was based on another referee's comment. But I think authors should change this description. For example, "Our results suggest _____ because some previous studies showed _______".

➢ We changed the phrases according to reviewer suggestions.

**List of changes**

➢ In order to follow the referees´ concerns that the title/statement of the manuscript is not justified we change it into: "Significance of climate and hydrochemistry …".

➢ We prepared revised figures in which the population form Punta Laguna is indicated.

➢ We extended the discussion to emphasize the novelty of the study in order to meet the referees´ critics that explanations are not persuasive and well-structured.

➢ We added a sentence at the end of the introduction to explicitly highlight the novelty of the study.

➢ We added some life cycle information in the revised discussion.

25  ➢ We changed the phrases "hypothesize" according to reviewer suggestions.

[revised manuscript text omitted]
 differentsmall geographical areas with high resolution and ranges and geographical scales. Accordingly, there are either relatively small regions studied with a high resolution (e.g., van der Meeren et al., 2010), or random larger study areasregions with widespread samplinge localities disregardingthat does not match the relation to the distribution of the species involved (e.g., Baltanas et al., 2002; Ramos et al., 2015; Boomer et al., 2017). Our approach is the first that covers a supra-regional scale that coincides with the geographical range of Cytheridellathe study taxon. Another novel contribution to the investigation of ecophenotypy represents the inclusion of climatic data in contrast to many other approaches whosewhere data-sets are often restricted to hydrochemical information. However, hHydrochemical conditions usually vary on small spatial scales in contrast to climatic data. Previous studies were therefore restricted to characterize local and orrather than regional interactionseffects betweenof environment and on shape changes. It is known from several other organism groups that species may exhibit differences in sensitivity to ecological conditions through their geographical ranges with a higher sensitivity of range-edge populations than those nearer to the center of the species' distribution (e.g., Mills et al., 2017). Thus, some environmental parameters that were identified in other studies as major control on morphological changes (such as salinity) could be important on more restricted geographical scales due to a higher sensitivity of the local populations. The ecophenotypical response to changes of environmental conditions could vary also with geographical scale. In contrast, oOur approach enables to investigate the overall pattern of ecophenotypical responses to environmental changes and minimizes local effects, which may have overemphasized in studies that consider only environmental variables of the water body and/or smaller geographical scales (cf. e.g., Ramos et al., 2015).
.

[revised manuscript text omitted]

---

## Author Response (AR3)

**Reply to Anonymous Referee #1 (Report submitted on 03 August 2018)**

We thank the referee for his/her continuing efforts to improve the manuscripts and constructive comments on the manuscript that helped to improve it.

The referee has only some minor comments which we answer point-by-point.

"Page 1, Lines 29-30: "and emphasize that environmental ... non-marine ostracods"
It is a little hard to follow the statement. It is recommended to remove the words or to add more information that the results contrast with expectation or previous studies. In the latter case, information noted in Lines 25-29 of Page 7 should be introduced in Introduction."

➢ We rephrased the respective sentence in order to remove confusions.

"Page 7, Lines 19-29, I recommend authors to move these information to section 4.1. in combination with Lines 17-27 of Page 8, authors can emphasize that
-Shpae variation is related to variation in temperature seasonality, annual precipitation and anions in this study
-These factors are unexpected from previous studies
-It is difficult to reveal the mechanism, but it is suggested to be attributed to physiological responses."

➢ The information were moved according to the suggestion of the reviewer.

"Page 11, Lines 16-18, Possible environmental similarity of Punta Laguna with Florida can be more discussed for emphasizing that the shape is linked to environment rather than geographic locations."

➢ We appreciate the intention of the referee. The presence of two co-occurring morphotypes in one location (Punta Laguna) indeed challenges the hypotheses that morphology is completely controlled by environmental factors. However, environmental similarity with, e.g, Floridian habitats, can be excluded because otherwise only one morphotype would occur. In order to avoid lengthy and very speculative discussion we prefer to confine this section to the statements that this co-occurrence could suggest the presence of microhabitats with specific environmental conditions, posing differential impact on valve calcification in the very same ecosystem. On the other hand, this

discrepancy might be considered evidence for genetic differentiation. An integrated study combining genetic and morphometric data is required to further explore this case.

5 "Fig. 4 I understand the role of PLS in this study, but I still recommend to add shape deformation along the axis. This is possible relatively easily for example by using geomorph package of R."

> We ran the analyses with the R package 'geomorph', but it does not provide the '% variance explained' per PLS axis we require. Also, it seems to use a slightly different algorithm because the results were slightly different. 10 Therefore, we would like to stick with our previous analyses. We anyway consider the shape deformations along the relative warps much more important.

**List of changes made in the manuscript**

- ➢ Work of the referees is thanked in the acknowledgements

5 ➢ Citation updated

- ➢ Changes made according to referees' suggestions

[revised manuscript text omitted]